# An Extension of the Failure Mode and Effect Analysis with Hesitant Fuzzy Sets to Assess the Occupational Hazards in the Construction Industry

**DOI:** 10.3390/ijerph17041442

**Published:** 2020-02-24

**Authors:** Jalil Heidary Dahooie, Amir Salar Vanaki, Hamid Reza Firoozfar, Edmundas Kazimieras Zavadskas, Audrius Čereška

**Affiliations:** 1Faculty of Management, University of Tehran, Jalal Al-e-Ahmad Ave., Nasr Bridge, Tehran 14155-6311, Iran; amirsalarvanaki@ut.ac.ir (A.S.V.); hmdfiroozfar@ut.ac.ir (H.R.F.); 2Institute of Sustainable Construction, Laboratory of Operational Research, Vilnius Gediminas Technical University, Sauletekio av. 11, Vilnius LT-10223, Lithuania; 3Department of Mechanical and Material Engineering, Vilnius Gediminas Technical University, Basanaviciaus str. 28, LT-03324 Vilnius, Lithuania; audrius.cereska@vgtu.lt

**Keywords:** construction industry, failure mode and effect analysis, hesitant fuzzy set, MCDM, occupational hazards, risk management

## Abstract

The construction industry is considered as one of the most dangerous industries in terms of occupational safety and has a high rate of occupational incidents and risks compared to other industries. Given the importance of identifying and assessing the occupational hazards in this industry, researchers have conducted numerous studies using statistical methods, multi-criteria decision-making methods, expert-based judgments, and so on. Although, these researchers have used linguistic variables, fuzzy sets and interval-valued intuitionistic fuzzy sets to overcome challenges such as uncertainty and ambiguity in the risk assessment conducted by experts; the previous models lack in efficiency if the experts are hesitant in their assessment. This leads to the inability to assign a specific membership degree to any risk. Therefore, in this research, it is tried to provide an improved approach to the Failure Mode and Effects Analysis (FMEA) method using an Multi-Criteria Decision-Making (MCDM) method based on the hesitant fuzzy set, which can effectively cope with the hesitance of the experts in the evaluation. Also, Stepwise Weight Assessment Ratio Analysis (SWARA) method is applied for risk factor weighing in the proposed approach. This model is applied to a construction industry case study to solve a realistic occupational risk assessment. Moreover, a comparison is made between the results of this model and those obtained by the conventional FMEA and some other aggregation operators. The results indicate that the newly developed approach is useful and flexible to address complex FMEA problems and can generate logical and reliable priority rankings for failure modes.

## 1. Introduction

The construction industry is considered as one of the infrastructure industries in different countries of the world and plays an important role in the development and growth of countries [1]. This sector is a labor-intensive industry that needs to use multiple electrical and mechanical equipment, but unfortunately, safety training for its workers is inadequate [2]. Hence, the construction industry has experienced a high rate of accidents and occupational injuries, so that it is currently among the most dangerous industries in the world. Evidence has shown that the fatality rate in the construction industry is at the highest level among the other industries in most of the world, including Europe, North America, and Australia, where it ranges from 3 to 14 work related death per 100,000 workers [3]. In 2014, the construction industry was responsible for more than 73,000 non-fatal injuries and approximately 20.5% of worker fatalities in the United States. Between 2003 and 2008, while merely about 7.8% of employees worked in the construction industry in New Zealand, over 12% of non-fatal injuries have occurred in this sector [4]. Occupational risks and hazards in this industry not only threaten people’s health but also economically challenge the organization by imposing a lot of costs on projects [1,5,6]. It has caused the main concerns of this area are to promote safety level and to counteract the occupational risks [7]. 

Given the limited resources of organizations, it is obvious that coping with all risks and hazards is virtually impossible. Thus, identifying and assessing risks and hazards in order to determine the most important ones is a crucial step in risk management [8]. This has led researchers to develop effective and efficient tools to identify, assess, and manage the occupational risks and hazards of this industry to cope with their health and financial implications maximally. Pinto et al. applied traditional methods and fuzzy set theoretical approaches to obtain a method for assessing occupational risks in construction sites [6]. Thomas et al. used the fuzzy-fault tree and the Delphi method to provide a framework for risk probability and impact assessment with the ability to evaluate the risks under inadequate data conditions [9]. Liu and Tsai presented a combined fuzzy-based risk assessment methodology to manage occupational risks and hazards of the construction industry. The approach presented in their study includes the combination of Quality Function Deployment (QFD), Analytical Network Process (ANP), and Failure Mode and Effects Analysis (FMEA) methods used to determine the relationships between construction items, types of risks and their causes, and finally, to assess each risk value [1]. Błazik-Borowa and Szer aimed at identifying the causes of the health hazards of workers in the construction industry and reducing their likelihood of the occurrence and analyzed the stages of scaffolding life used in the construction projects. They investigated different stages of scaffold design and usage to identify failures and errors in each step. They also have suggested ways to reduce the probability of the occurrence of these risks [10]. Seker and Zavadskas presented a model using a fuzzy-based DEMATEL (Decision making trial and evaluation laboratory) method to assess the occupational hazards of the construction industry. One of the main advantages of the model presented in this study is to consider causal relationships in determining the importance of risks [2]. Debnath and Biswas developed a model for assessing the health risks of construction workers using the interval type-2 fuzzy analytic hierarchy process [11]. 

In general, the risk assessment problem can be considered as a multi-criteria decision-making problem. Various researchers have suggested that multi-criteria decision-making methods be used to solve these problems due to their high ability to assess and prioritize [12,13,14]. In many of the studies conducted on risk assessment, researchers have used MCDM methods as well as expert knowledge to assess the risks. Also, due to the uncertain nature of risk, fuzzy sets [1,2], interval-valued fuzzy sets [11,15], interval-valued intuitionistic fuzzy sets [16,17], and so on are used in many models to overcome this kind of uncertainty and ambiguity. On the other hand, in many real-world problems in the field of risk assessment, for various reasons such as differences in experience and knowledge level of experts, time pressures, etc., decision-makers do not usually come to a consensus on complex decisions on specific elements and getting an agreement is usually difficult. For example, there may be differences between the opinions of two experts on the membership degree of the element *x* in the set A (one expert determines the value of 0.4, and the other expert determines the value of 0.8 for membership degree). Also, each expert may be individually skeptical about determining the membership degree of an element in a particular set and expresses different values for membership degree. These conditions cause decision-makers to come up with a set of possible values for the membership degree instead of a single and distinct membership degree value. 

In order to overcome these challenges and eliminate the complications arising from them in the decision-making process, a concept called hesitant fuzzy sets was presented by Torra and Narukawa as a development of a fuzzy set [18]. A hesitant fuzzy set provides the possibility to consider a set of possible values between [0,1] as a membership function, which greatly increases its effectiveness in solving real-world problems. Since the emergence of the hesitant fuzzy concept, the growing attention of scholars has come to this area, and it has been successfully used in many decision-making problems under the conditions of uncertainty [19,20,21].

Given the above, the main goal of this paper is to present an extended FMEA method, using the MCDM method based on the hesitant fuzzy numbers and the SWARA approach, not only to overcome the aforementioned deficiencies of traditional FMEA but also to have a flexible risk assessment tool, which effectively considers the hesitancy of FMEA team members, by taking the advantages of hesitant fuzzy numbers. Moreover, applying the presented approach in the construction industry to determine the most important occupational hazards in this sector could be considered as the minor goal of this study. To do so, a hybrid model is presented in this study in which the SWARA method, the hesitant fuzzy-based Multiple-Criteria Decision Making (MCDM) method, and the FMEA method are used to provide a more accurate assessment of the occupational risks in the construction industry. FMEA is a systematic tool for identifying and assessing potential failures. It was developed by the U.S. National Aeronautics and Space Administration (NASA) for the first time in the early 1960s to improve the reliability of military equipment [22]. Among the important features of this technique is an action-oriented approach to the risks which emphasizes risk prevention rather than providing solutions to the problem [23,24]. Although there are criticisms about the FMEA method, this technique has attracted enormous interest and has become a widely used tool in risk assessment due to its easiness and high visibility [25]. Given that the risk assessment using the FMEA method is a group decision based on the views of the members of the FMEA team, so due to the differences in the level of experience, knowledge, and expertise, it is quite likely that the members of the FMEA team have different views on risk assessment or are hesitant about estimating a risk and evaluating it accurately. This issue distorts the consensus of the opinions of team members. As the main innovation of this research, the model presented in this paper can solve this problem by hesitant fuzzy logic to increase the accuracy of risk assessment. Also, to further enhance the accuracy of the assessment, the risk weight of the factors is obtained using the Stepwise Weight Assessment Ratio Analysis (SWARA) method, which is one of the most widely used MCDM methods to determine the importance of the criteria. This method helps coordinate and gather data from experts [26].

## 2. Literature Review

FMEA is a powerful team-driven tool for assessing the security and stability of products, services, processes, and systems that are designed to define, identify, and eliminate known or potential failures, problems, and errors [24,27]. It is an appropriate risk assessment tool that helps decision-makers and risk analysts identify potential failure modes to rank them in order of importance so that measures are taken to reduce the probability of occurrence and the risks posed by the riskiest failure modes [28]. Regarding the strengths of this approach in risk assessment, it widely has been applied by researchers in different industries like marine, aerospace, automobile, healthcare, and so on to assess and prioritize the risks [29]. Also, it has been used to solve the risk assessment problems in the construction industry, which crucially needs a powerful risk assessment tool because of the complexity and high level of risks in the projects [30].

The risk assessment in this technique is based on three criteria: Severity (S), Occurrence (O), and Detection (D) [8]. In the traditional FMEA, to measure the three criteria, a numeric scale of 1 to 10 is used. Thus, each failure mode is characterized by three numbers, and by multiplying them together, a scale called Risk Priority Number (RPN) is obtained for ranking the failure modes [31]. Accordingly, the higher the RPN value, the more important the failure mode is, and the higher priority it has.

Although FMEA is widely used in risk management, many criticisms have accompanied its traditional form through previous researches. The most serious criticisms include [8,27,28,32,33,34]:The equal weights for O, S, D may not occur in practice, and these parameters actually have different weights.Different combinations of the three parameters may result in the production of the same RPN number for two or more failures, while their potential risks may be completely different. For example, two different failure modes that have O, S, and D values of 1, 10, 8, and 4, 5, 4, respectively, both have the same RPN value of 80 and are of the equal importance and priority in the traditional FMEA method, but in practice, they may be very different.It is difficult to assess the exact value of risk factors because it is always faced with uncertainties due to the low time of FMEA team members, their lack of full control over all aspects, personal opinions, and so on.There is no mathematical justification for multiplying the number of risk factors together to obtain RPN.Numbers assigned to risk factors are of a sequential scale, so their multiplication is not only meaningless but even misleading.RPNs are not continuous and heavily distributed at the bottom of the scale from 1 to 1000. Many of the numbers in the range of 1–1000 cannot be formed from the product of O, S, and D, and only 120 of the 1000 numbers are unique.The definition of the risk index is affected by the expert’s attitude toward risk. Moreover, group decision-making is not considered in the traditional FMEA method.

Given the criticisms of the traditional FMEA method, a lot of efforts have been made in the literature to improve the performance of this method and to resolve these criticisms. Since determining the rank and priority of failure modes in the FMEA method is an MCDM problem [31], many researchers have used these methods in improving the FMEA method. Also, various methods such as fuzzy set, interval-valued intuitionistic fuzzy set, grey numbers are used to overcome the uncertainties in the risk assessment. The following are some of these studies.

Bowles and Pelaez were the first to use fuzzy logic to overcome the uncertainties in the assessment of failure modes. In their model, the fuzzy method transforms the risk factors into fuzzy representations, which are complied with the rules in a fuzzy if-then rule base and defuzzified in order to determine the risk degrees of failure modes [35]. Ko used a two-tuple linguistic model to assess failure modes for averting the information loss in the FMEA process [36]. Liu et al. presented a model using a combined approach, D numbers and the Grey Relational Project (GRP), to improve the FMEA method for evaluation under conditions of uncertainty. In this approach, risk factors are evaluated using D numbers, and the priority of risks is calculated using the GRP method [37]. Wang et al. (2016) used the combined ANP and Complex Proportional Assessment (COPRAS) methods with interval-valued intuitionistic fuzzy numbers to solve the traditional FMEA method problems under the conditions of uncertainty and its inefficiency in determining the weights of criteria and priorities of risks [17]. Liu et al. used hesitant fuzzy linguistic variables to evaluate failure modes in order to better aggregate the views of the members of the FMEA team and perform a more accurate evaluation and applied the Qualitative Flexible Multiple Criteria Method (QUALIFLEX) method to rank them [25]. Liu et al. proposed a model to overcome the limitations of the traditional FMEA methodology. In the model presented in their research, the cloud model theory is used to represent the uncertain linguistic evaluations given by members of the FMEA team, and the extended GRA method is used to determine the priority of failure modes. Also, in this model, the relative importance of each of the risk factors is calculated based on GRA idea using a multi-objective optimization model [38]. Tsai et al. used the DEMATEL method to increase the effectiveness of the traditional FMEA method. In their model, FMEA was applied to identify the items needed to improve. Then, DEMATEL was used to investigate the interactive effects and causal relationships of these items. In the final step, solutions for problems were prioritized [39]. Fattahi and Khalilzadeh developed an extended methodology to improve the accuracy of the FMEA method. In their method, the Fuzzy Weighted Risk Priority Number (FWRPN) was used instead of RPN for each failure. The extended fuzzy Analytical Hierarchy Process (AHP) and Fuzzy Multi-Objective Optimization Based on Ratio Analysis (MULTIMOORA) plus the Full Multiplicative Form methods were applied to calculate the weights of three factors and the weights of failure modes, respectively [40]. Yousefi et al. aimed at addressing the problems and inadequacies of the FMEA prioritization system and developed a new FMEA model using the Data Envelopment Analysis (DEA) method [41]. Bian et al. used the combination of D number and Technique for the Order of Preference by Similarity to Ideal Solution (TOPSIS) to improve FMEA. They proposed a novel risk priority model based on D numbers and TOPSIS for risk assessment in FMEA. The assessments performed by the FMEA team members were denoted by D numbers, where a novel practical and efficient method was able to represent the uncertain information effectively [42]. Geramian et al. used fuzzy logic to overcome the problems of group thinking in the process of risk assessment using the FMEA method. In the model developed by these researchers, the Taguchi’s robust parameter design was used, and the effects of various control parameters, including defuzzification, Aggregation, and Implication operators for the fuzzy inference system (FIS) were investigated [43]. In addition to the above researches, in order to improve the traditional FMEA methodology and fix its challenges and limitations, other models and methods have been proposed by the researchers, including models based on TOPSIS [44], DEA [45], DEMATEL [34], and so on. 

Liu et al., in a review paper, asserted that in spite of the several attempts that made to improve the traditional FMEA and overcome its shortcomings, the hesitancy of decision-makers in risk assessment processes remains as a serious problem which needs to be solved by using appropriate methods and operators [29]. Particularly, in the construction industry, the risk assessment is significantly more complicated than the other sectors because of the high complexity of the construction projects. Hence, using the hesitant fuzzy sets is more consistent with this type of complicated projects [46].

Therefore, to overcome the mentioned shortcomings of traditional FMEA and solve the problem of the hesitancy of decision-makers in the risk assessment process, in this paper, we proposed a new hybrid FMEA method using the SWARA method and the HFWGHM operator. Finally, the validity of the results of this method is analyzed.

## 3. The Proposed Hesitant Fuzzy FMEA Method

The risk assessment using the model presented in this paper consists of three phases, as shown in Figure 1. This model is based on the use of the SWARA method and HFWGHM-based MCDM method in a hesitant fuzzy environment. The phases of the model presented in this study are described below.


*Phase 1:*


At this stage of the risk assessment process using the proposed model, the FMEA team consists of experts and decision-makers specializing in the subject is formed, and the list of potential failure modes is determined and finalized by team members. At the final step of the first phase, the failures identified by the experts are evaluated based on risk factors of the FMEA method using linguistic variables, and the initial decision matrix is formed by linguistic variables. Regarding what has already been said about the model presented in this research, it is evident that each member of the FMEA team presents his own personal and expert opinions. Hence, their doubts and ambiguities in the assessment of failure modes based on the risk factors have been considered.


*Phase 2:*


At this stage of the process, the weights of risk factors are assessed by the FMEA team members separately based on the SWARA method. The weight of each risk factor is calculated by the SWARA method, and the final weights are determined using the geometric mean of the weights obtained from the opinions of the FMEA team members.


*Phase 3:*


In order to assess the failure modes and rank them using the proposed model, it is necessary to follow the following steps that are derived from [47]:


*Step 1.*


If $=\left\{FM_{1},FM_{2},\ldots ,FM_{m}\right\}$ is a set of failure modes and C={O,S,D} is a set of risk factors and $\omega ={\left(\omega_{1},\omega_{2},\omega_{3}\right)}^{T}$ is the corresponding weight vector such that $\omega_{i}>0$ and $\overset{3}{\underset{i=1}{\sum }}\omega_{i}=1$; then, all possible values for evaluation of the failure mode $FM_{i}$ based on the risk factor $c_{i}$ (O, S or D) as an HFE are in the form of $h_{ij}=\cup_{\gamma_{ij}\in h_{ij}}^{}\left\{\gamma_{ij}\right\}$, and the set of these values for all $h_{ij}\left(i=1,2,\ldots ,m,j=1,2,3\right)$ forms the hesitant fuzzy decision matrix. This matrix is represented by $H={\left(h_{ij}\right)}_{m\times 3}$ which is shown in Table 1.


*Step 2.*


The second step is to make possible the comparison between elements of the decision matrix $H={\left(h_{ij}\right)}_{m\times 3}$. Hence, this matrix is normalized. The normalized hesitant fuzzy decision matrix is represented by $B={\left(b_{ij}\right)}_{m\times 3}$ and is defined as Equation (1). In this equation, $\cup_{\gamma_{ij}\in h_{ij}}^{}\left\{\gamma_{ij}\right\}=h^{c}$⋃ is complementary of h:
(1)$$b_{ij}=\cup t_{ij}\in b_{ij}=\{\begin{array}{c} \left\{\gamma_{ij}\right\}\;for\;benefit\;criterion\;c_{ij}\\ \left\{1-\gamma_{ij}\right\}for\;cost\;criteron\;c_{ij} \end{array}\;i=1,2,\ldots ,m;j=1,2,3$$


*Step 3.*


The HFWGHM operator, explained in the next section, is applied to the i-th row to aggregate the performance values $b_{ij}\left(j=1,2,3\right)$, and the total performance value corresponding to the failure mode $FM_{i}$, represented by $b_{i}$, is calculated using Equation (2):(2)$$b_{i}=HFWGHM\left(b_{i1},b_{i2},b_{i3}\right)$$


*Step 4.*


At this step, the score function $s\left(b_{i}\right)$ corresponding to $b_{i}$ is calculated using the Definition 3 and the failure modes are ranked based on $s\left(b_{i}\right)$ values in descending order.

### 3.1. Hesitant Fuzzy Weighted Geometric Heronian Mean (HFWGHM)

The concept of the hesitant fuzzy sets was presented by Torra and Narukawa and Torra, in which the membership values can be represented as a set of possible values [18,48]. So far, several decision-making models based on a hesitant fuzzy set are proposed in various studies to make a decision under uncertainty, such as [49,50].

**Definition** **1.***Let*X*be a fixed set, then the hesitant fuzzy set on*X*is defined in terms of a function, when applied to the X set, returns a subset of numbers in*[0,1]*. This set can be displayed as follows [48]:*(3)E={〈x,h(x)〉|x∈X}*where* h(x)*is a set of values in*[0,1]*, which indicates the possible membership degree of each*x∈X*in the set E. For ease, HFE (Hesitant Fuzzy Element) is used to refer to*h(x).

**Definition** **2.**
*If h,*
$h_{1}$
*, and*
$h_{2}$
*are three HFEs then:*
(1)$\;h^{\lambda }=\cup_{\gamma \in h}\left\{\gamma^{\lambda }\right\}$;(2)$\lambda h=\cup_{\gamma \in h}\left\{1-{\left(1-\gamma \right)}^{\lambda }\right\}$;(3)$h_{1}⨁h_{2}=\cup_{\gamma_{1}\in h_{1},\gamma_{2}\in h_{2}}\left\{\gamma_{1}+\gamma_{2}-\gamma_{1}\gamma_{2}\right\}$;(4)$h_{1}⨂h_{2}=\cup_{\gamma_{1}\in h_{1},\gamma_{2}\in h_{2}}\left\{\gamma_{1}\gamma_{2}\right\}$.


**Definition** **3.**
*For hesitant fuzzy element h,*
$s\left(h\right)=\frac{1}{\#h}\underset{\gamma \in h}{\sum }\gamma$
*is called the score function, where*
#h
*represents the number of elements of the set h. In addition, if h,*
$h_{1}$
*, and*
$h_{2}$
*are three HFEs, then the following rules and relationships govern this set [51]:*
(1)*If*$s\left(h_{1}\right)>s\left(h_{2}\right)$*then,*$h_{1}>h_{2}$;(2)*If*$s\left(h_{1}\right)=s\left(h_{2}\right)$*then,*$h_{1}=h_{2}$. The Hesitant Fuzzy Weighted Geometric Heronian Mean (HFWGHM) was developed by [47].


**Definition** **4.**
*Consider*
$h_{i}\left(i=1,2,\ldots ,n\right)$
*to be a set of HFEs and*
ω=(ω1,ω2,…,ωn)T
*to be the weight vector of*
hi
*, where*
ωi
*represents the importance degree of*
$h_{i}$
*such that*
$\omega_{i}>0$
*and*
$\overset{n}{\underset{i=1}{\sum }}\omega_{i}=1$
*. Then the weighted operators, Hesitant Fuzzy Weighted Geometric Heronian Mean (HFWHM) and HFWGHM, are defined as Equations (4) and (5):*
(4)$$HFWHM\left(h_{1},h_{2},\ldots ,h_{n}\right)={\left(\frac{2}{n\left(n+1\right)}\overset{n}{\underset{i=1}{\sum }}\overset{n}{\underset{j=i}{\sum }}{\left(w_{i}h_{i}\right)}^{p}⨂{\left(w_{j}h_{j}\right)}^{q}\right)}^{\frac{1}{p+q}}$$
(5)$$HFWGHM\left(h_{1},h_{2},\ldots ,h_{n}\right)=\frac{1}{p+q}\left(\begin{array}{c} n\\ ⨂\\ i=1,j=i \end{array}{\left({\left(ph_{i}\right)}^{w_{i}}⨁{\left(qh_{j}\right)}^{w_{j}}\right)}^{\frac{2}{n\left(n+1\right)}}\right)$$


**Theorem** **1.**
*Consider*
$h_{i}\left(i=1,2,\ldots ,n\right)$
*to be a set of HFEs and*
ω=(ω1,ω2,…,ωn)T
*to be the weight vector of*
hi
*, where*
ωi
*represents the importance degree of*
$h_{i}$
*, such that*
$\omega_{i}>0$
*and*
$\overset{n}{\underset{i=1}{\sum }}\omega_{i}=1$
*. The aggregated value obtained from the HFWGHM or HFWHM operators will be in the form of Equations (6) and (7), respectively. The proof of this theorem is presented in [47]:*
(6)$$HFWHM\left(h_{1},h_{2},\ldots ,h_{n}\right)=\;\underset{\xi_{i}\in h_{i},\xi_{j}\in h_{j}}{\cup }\left\{{\left(1\overset{n}{\underset{i=1,j=i}{\prod }}{\left(1{\left(1-{\left(1-\xi_{i}\right)}^{w_{i}}\right)}^{p}{\left(1-{\left(1-\xi_{j}\right)}^{w_{j}}\right)}^{q}\right)}^{\frac{2}{n\left(n+1\right)}}\right)}^{\frac{1}{p+q}}\right\}$$
(7)$$HFWGHM\left(h_{1},h_{2},\ldots ,h_{n}\right)=\;\underset{\xi_{i}\in h_{i},\xi_{j}\in h_{j}}{\cup }\left\{1-{\left(1-\overset{n}{\underset{i=1,j=i}{\prod }}{\left(1-{\left(1-\xi_{i}{}^{w_{i}}\right)}^{p}{\left(1-\xi_{j}{}^{w_{j}}\right)}^{q}\right)}^{\frac{2}{n\left(n+1\right)}}\right)}^{\frac{1}{p+q}}\right\}$$


Due to the capabilities of HFSs, they have been used to resolve many practical decision-making problems such as contractor selection in logistics outsourcing [52], investment opportunity selection [53] and ranking engineering characteristics in product development of electric vehicles [54].

### 3.2. SWARA Method

Experts have an important role in evaluating and determining the weight of the criteria in multi-criteria decision-making problems. They are in charge of the inescapable part of the decision-making process. Each expert identifies the priority of each criterion, and ultimately, the criteria ranks are obtained according to the overall output of the priority settings. The most important factors leading to better decision making by experts are knowledge, information, and experience. In the SWARA method, as a new MCDM method, the highest rank indicates the most valuable criterion, while the lowest rank shows the least important criteria. After that, the geometric mean of ranking values is taken to determine the final ranking [55]. The capability of the SWARA method to evaluate expert precision about the criteria is considered to be its major advantage [56]. The point of view of SWARA differs from other MCDM approaches, such as ANP, Factor Relationship (FARE), and the AHP [57]. This approach leads to improved decision making in a wide range of situations and a more appropriate ranking of the criteria for necessary goals. In addition, expert opinions are applied to the decision-making process.

SWARA provides the following benefits compared to other weighting tools and methods; it handles the ability for estimation of experts’ opinion about criteria importance ratio in determining the weights, it has the benefit of coordinating and gathering information from experts, it is user-friendly, uncomplicated, and simple, in which the experts have the ability to easily work together, and finally, it explores the problem priorities according to company policies [58].

The SWARA method has been applied to various decision-making problems in some previous studies [59,60,61]. Below is a summary of the main steps of this method:

*Step 1*: Sorting criteria based on expert opinions. The most important criterion is ranked first, and less important criteria are in the next levels.

*Step 2*: Determining the comparative importance of each criterion. The relative importance of each criterion is determined in relation to previous criteria. This comparative importance is denoted by Sj.

*Step 3*: Calculating Kj, a relative importance function for each criterion. This coefficient is calculated using Equation (8):(8)$$K_{j}=S_{j}+1$$

*Step 4*: Calculating the initial weight of each criterion, which can be determined using Equation (9):(9)$$w_{j}=\frac{w_{j-1}}{K_{j}}$$

*Step 5*: Determining the final weight of each criterion. This is the final step in the SWARA method, in which the final weight of each criterion, the normalized weight, is obtained using Equation (10):(10)$$q_{j}=\frac{w_{j}}{\sum w_{j}}$$

## 4. Illustrative Example

In this section, we provide a case study to illustrate the potential applications and benefits of the proposed FMEA in the prioritization of failure modes. For this purpose, the model presented in this study is used to assess the occupational risks of the construction industry. 

The case study is about an active company in various fields of designing and supervising the implementation of construction and urbanization projects in Iran. Faratarh Ariana Consulting Engineers Company has been active in the construction industry for more than a decade and has been designer and supervisor of more than four million square meters of building in the form of various projects. The company’s high expertise in consulting and supervising construction projects, and the plurality of projects distributed throughout Iran and sometimes abroad, have caused the company’s managers and experts to have a broad perspective on the occupational hazards in the construction industry and to be able to assess the risks accurately. This is why this company has been selected as the case study.

### Implementation

Based on the process outlined in Figure 1, after identifying the risk assessment objectives and defining the level of analysis, a FMEA team comprised of 4 company managers was formed, and the occupational hazards of the construction industry were identified as failure modes. After the consensus of the team members’ opinions, the failure modes were finalized, as shown in Table 2.

At this stage, FMEA team members were asked to separately express their opinions about the relative importance of risk factors based on the SWARA method steps. For example, the opinions of one of them in the risk weighting process are shown in Table 3.

The final weight of each of the risk factor were determined by calculating the geometric mean of the weights determined by each member of the FMEA team. These values are shown in Table 4.

Also, at this stage, the experts were asked to assess the identified failure modes using the linguistic variables specified in Table 5. The results of this assessment, known as the initial decision matrix, are presented in Table 6.

Then, using Equation (4), the assessments conducted by experts were converted into the hesitant fuzzy set. In other words, the initial decision matrix changed to the hesitant fuzzy decision matrix. This matrix is shown in Table 7. Also, by converting the linguistic variables into fuzzy numbers, the numeric form of the hesitant fuzzy decision matrix was formed as Table 8.

As can be seen in this matrix, the difference between the expert opinions in the assessments has been well-modeled. For example, in the above decision matrix, consider $h_{21}=\left\{0.9\right\}$. This means that the experts had the same view in the assessment of the alternative FM2 based on the Occurrence criterion. As another example, consider the value $h_{33}=\left\{0.1,\;0.7,\;0.9\right\}$. This means that two of the four members of the FMEA team had the same view in the FM3 assessment based on the Detection risk factor, and the other two members had different opinions. Therefore, the HFE for FM3 assessment based on the Detection criterion has three different members.

Now, based on the second step of the hesitant fuzzy MCDM, the specified hesitant fuzzy decision matrix should be normalized using Equation (1). It is worth noting that since Detection risk factor has a different nature compared to the other two risk factors, the more the values of Detection, the less the importance of the risk. 

In the traditional FMEA method, if the value of the detection for a failure mode is greater, then the lower number is considered for it, which ultimately results in lower RPN and decreasing the importance of the failure mode, but in the logic considered in this research, the higher detection value is evaluated with more linguistic variables, and when this criterion is calculated, it is considered as a negative criterion, which actually reduces the importance of the failure mode, so, given that the Detection criterion is considered as a negative criterion and according to Equation (1), the complementary of HFEs should be calculated for this criterion to normalize the hesitant fuzzy decision matrix. This matrix is presented in Table 9.

In this step, the risk values of each failure mode were aggregated using the HFWGHM operator, and the total risk values corresponding to each failure mode were calculated using Equation (2). After calculating the overall performance values of each failure mode, the performance score of each failure mode was calculated using the Equation (5), and then, the alternatives were ranked based on these scores. 

## 5. Results and Discussion

The performance rating and ranking have been calculated by the HFWGHM operator (p *=* q = 1) and presented in Table 10.

Given the final results, the failure modes can be sorted in descending order: FM2 > FM4 > FM5 > FM1 > FM8 > FM7 > FM3 > FM9 > FM6. It is clear that the failure mode FM2 suffers a high-risk degree, and it should be emphasized and regarded meticulously. The achieved results were shared with security and safety experts in the field of study, and they confirmed such results. Hence, it can be concluded that the proposed model is reliable and informative.

In order to evaluate the rationality and reliability associated with the proposed approach of failure mode ranking, the authors conducted sensitivity analyses for the current approach and compared it with some results of other methods.

### 5.1. Sensitivity Study

Equation (5) denotes that in the suggested procedure, two p and q parameters are used to calculate the score and rank of the failure modes. In the real decision-making problems, the decision-makers taking a gloomy view of the prospects can select the higher p and q, while optimistic decision-makers can consider lower values for them [62]. In this case, p and q are considered values equal to 1. However, both parameters can accept any non-negative integers. In order to show the effect of the parameters p and q in this problem, the performance scores of failure modes were calculated based on different values of p and q, and the alternatives were ranked considering each of these states. The results are shown in Table 11.

In addition, in the case where the value of one parameter (p or q) is considered constant, and the other parameter is continuously changed, the trends of changes in the performance score of each failure mode are shown in Figure 2 and Figure 3.

As Table 11 and Figure 2 and Figure 3 indicate, the final ranking of the failure modes can vary according to p and q. Precisely, FM2, failing objects, in all the scenarios, has acquired the highest priority, and also FM6, structure collapse, has taken the lowest priority under different values for p and q. Thus, the decision-making model based on the HFWGHM operator for these two failure modes presents excellent stability. Whereas, with increasing p and q values to the values greater than 1, the order of middle failure modes changes slightly (in some cases, they swapped with each other), which demonstrates the influence of the features of decision-makers (their risk-taking and risk-aversion behavior) in decision-making processes.

Changes in the values of the two parameters p and q, also result in changes in the aggregated values (using the HFWGHM operator) of HFEs, as shown in Figure 4, separated for each of the alternatives. The value and range assessment of the diagrams pertinent to the performance rating of the different options (Figure 4) illustrates that FM2 is the most significant failure mode, and FM6 is the most insignificant failure mode.

### 5.2. Comparison Analysis

Furthermore, in order to analyze the effectiveness of the presented method, the results should be compared with other extended FMEA methods. But, because of the important role of p and q parameters in the final ranking, the results are compared with other operators in the scope of hesitant fuzzy numbers like hesitant fuzzy weighted Heronian mean operator (HFWHM), the hesitant fuzzy weighted Bonferroni mean (HFWBM) [63] and hesitant fuzzy weighted geometric Bonferroni means (HFWGBM) [64] operators. The final ranking of alternatives, for different values of p and q, is presented in Table 12.

As shown in Table 12, the rank assigned to each of the failure modes using the traditional FMEA method in some cases, such as FM1 and FM4, is completely different from the rank obtained by the MCDM method based on different operators of the hesitant fuzzy numbers, which is due to the limitations of the traditional FMEA method described in Section 2. For example, in the traditional FMEA method, FM1 has a higher priority and importance than FM2, while in the logic of hesitant fuzzy numbers, and in particular in the model presented in this study, for all values of p and q, FM2 has a higher priority than FM1, which is more compatible with real-world events according to experts’ opinion. One of the main reasons for this difference is the consideration of different weights for risk factors in the model presented in this study. Accordingly, Occurrence has a higher weight than Severity, and since FM2 is more common than FM1, it has a higher priority. Also, the hesitant fuzzy numbers have been used in the model presented in this study, and the difference in opinions between the members of the FMEA team has been carefully applied in the final aggregation; so these may also be other reasons for the difference in ranks of FMs. It is also observed that some of the failure modes (FM3 and FM6) in the traditional FMEA method have the same RPN and, consequently, the same importance, while in the model presented in this study, i.e., the MCDM method based on HFWGHM operator, FM3 placed 2-3 ranks higher than FM6 for different values of p and q. 

As discussed earlier, in the presented method, the values of p and q parameters play a crucial role in the decision-making process. Hence, in order to eliminate the effect of these parameters in results and prioritization and make a proper comparison between the proposed method and the other operators, they are assumed equal values for all calculations. The comparison between the proposed FMEA method and the other operators are depicted in Figure 5.

Moreover, to have a more straightforward comparison, the correlation coefficients between the ranking results of all approaches are calculated and shown in Table 13. Accordingly, there is a much significant correlation between the ranking resulted from HFWGHM and HFWGBM approaches, which are based on geometric mean, for all values of p and q parameters. Moreover, FM2 always stands in the first place of ranking order using these approaches. On the other hand, the other operators, HFWHM and HFWBM, are relatively correlated, and FM4 acquired the first rank in most situations using these two operators.

The potential causes of difference among the proposed method and the conventional FMEA method are as follows: (1) The proposed method can consider different types of uncertainty in assessments performed by the FMEA team, including imprecise, uncertainty, and hesitation. Practically, the accuracy of the opinions of FMEA team members improves because they are more flexible and able to express their mental opinions. It can be said that the main difference of the model presented in this study with other improved FMEA models is that the opinions of all members of the FMEA team are accurately applied in the calculation process. Perhaps it’s better to say that, in the proposed approach, the FMEA team members have the ability to provide different sets of values for evaluating failure modes. (2) In the traditional FMEA, all risk factors are considered equally important, while in the presented method, risk factors are weighted using the SWARA method, which is one of the well-known weighing methods and is helpful for coordinating and gathering information from FMEA team members. This weighing method is completely in contrast to the traditional FMEA method but improves the results by increasing their realism. 

Moreover, the presented approach has multiple advantages over conventional FMEA. For example, duplicated risk rankings are less likely to occur, while in the traditional risk assessment, its occurrence probability is high. The presented approach has also resolved other common limitations and criticisms in the traditional FMEA method, including the lack of mathematical logic in computing RPN.

As discussed earlier, according to experts’ opinion, the ranking results of the proposed method in this paper are more consistent with the real-world situation. Moreover, analyzing the difference between the results of the proposed approach and the other operators in the realm of hesitant fuzzy numbers has illustrated that using the various aggregation and meaning algorithms is the main cause of such difference. The hesitant fuzzy MCDM method based on the HFWGHM operator uses the geometric mean operator for aggregation of the opinions of various experts and group decision-making. On the other hand, given that the Heronian mean (HM) can consider the interrelationship between the individual criterion, it leads to relieving the calculation redundancy more than the methods which use Bonferroni mean (BM) [65]. Moreover, the results have shown that the different values of p and q parameters lead to different ranking orders. Hence, the features of the experts, their risk-taking and risk-aversion, can be considered in the risk assessment process by adjusting the values of p and q parameters. Such a capability makes the proposed method more flexible than the previous hybrid FMEA-MADM methods.

## 6. Conclusions

The construction industry has always been recognized as one of the most dangerous industries in the world, which is prone to reveal a wide range of occupational accidents. This has led various researchers to develop occupational risk assessment models to use in this industry. Several studies used the conventional FMEA method in the construction industry, but it has several shortcomings that compelled researchers to extend the FMEA using different MCDM methods. Although those hybrid methods have significant advantages over traditional FMEA, there are still some problems that need to get improved.

Given the uncertainty existing in risk assessment, in most of the proposed models, researchers have used the logic of fuzzy and intuitive fuzzy sets to increase the accuracy of the assessment and to overcome the existing uncertainty. However, if experts and evaluators are skeptical about risk assessment, the logic of fuzzy sets and intuitive fuzzy sets will not be effective. For a variety of reasons, including the difference in the level of knowledge and experience, the type and manner of looking at the problem and the level of optimism and pessimism of the experts, the skepticism of experts in risk assessment is likely to be quite feasible. In this situation, in order to overcome this uncertainty and hesitance, the use of hesitant fuzzy set will be helpful. Therefore, in this research, the SWARA method and MCDM method based on hesitant fuzzy logic are used to develop a new FMEA framework for assessing the occupational hazards of the construction industry which in addition to high ability to assess risk and overcome information uncertainty and expert hesitance, it also resolves deficiencies and difficulties of the FMEA framework. This model was applied to the occupational risk assessment problem in the construction industry, and its results were compared with traditional FMEA method and other hesitant fuzzy operators, including HFWHM, HFWBM, and HFWGBM. According to this comparison, the high accuracy and efficiency of the proposed model were proven for aggregation of the opinions of FMEA team members and the assessment of occupational hazards. The new model also has the ability to handle the information diversity and uncertainty in the FMEA team member’s assessment using the hesitant fuzzy numbers. In particular, the proposed FMEA framework is helpful for addressing complicated risk analysis problems, which include comprehensive risk factors and limited failure mode. Also, one of the limitations of the traditional FMEA methodology is the allocation of the same weight to risk factors, which reduces the accuracy of risk assessment and has made a lot of criticism. In the model presented in this study, this limitation has been fixed using the SWARA method to determine the weight of risk factors. In addition to simplicity and ease of use, this method has a high degree of accuracy in explaining the views of the members of the FMEA team, which greatly increases the accuracy of the assessment.

Given the above, the proposed approach in this paper has several advantages over than the conventional FMEA, including:By taking the benefits of the hesitant fuzzy sets, modeling the hesitancy of experts in the risk assessment process is made possible, and the effectiveness of their decisions is increased.By using the SWARA method, different weights could be considered for the risk factors, which leads to more accurate resultsBy using the HFWGHM-based MCDM, the problem of multiplication of values of the risk factors, as one of the mathematical problems of the traditional FMEA, is solved.The developed approach can provide possibilities more logically to regard FMEA team members’ opinion and group decision-making to prioritize risks.

In order to conduct further research in this area, it is suggested that the approach developed in this paper is adopted to evaluate risk in a variety of areas, such as technology management and human resource management, which have hesitant and uncertain data. Also, it is proposed to use a hybrid approach, hesitant fuzzy multi-attribute decision-making, and FMEA, to risk assessment problems.

## Figures and Tables

**Figure 1 ijerph-17-01442-f001:**
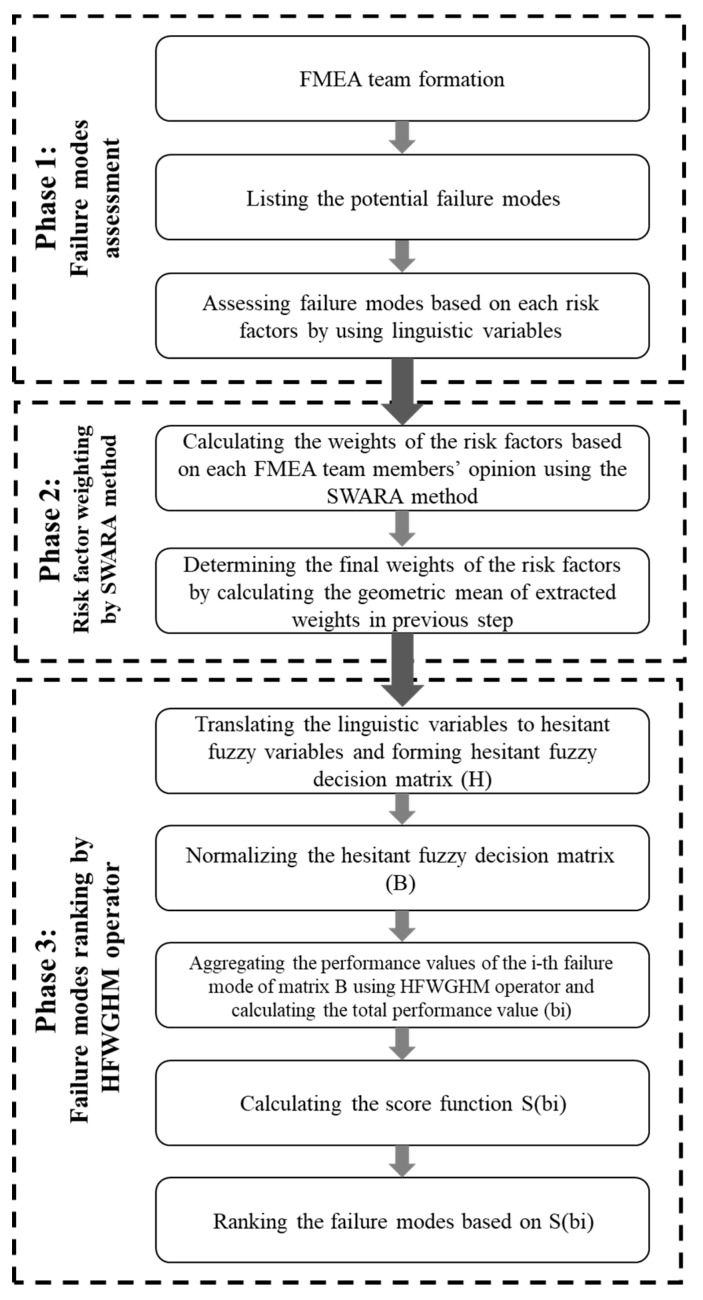
Framework of the proposed FMEA.

**Figure 2 ijerph-17-01442-f002:**
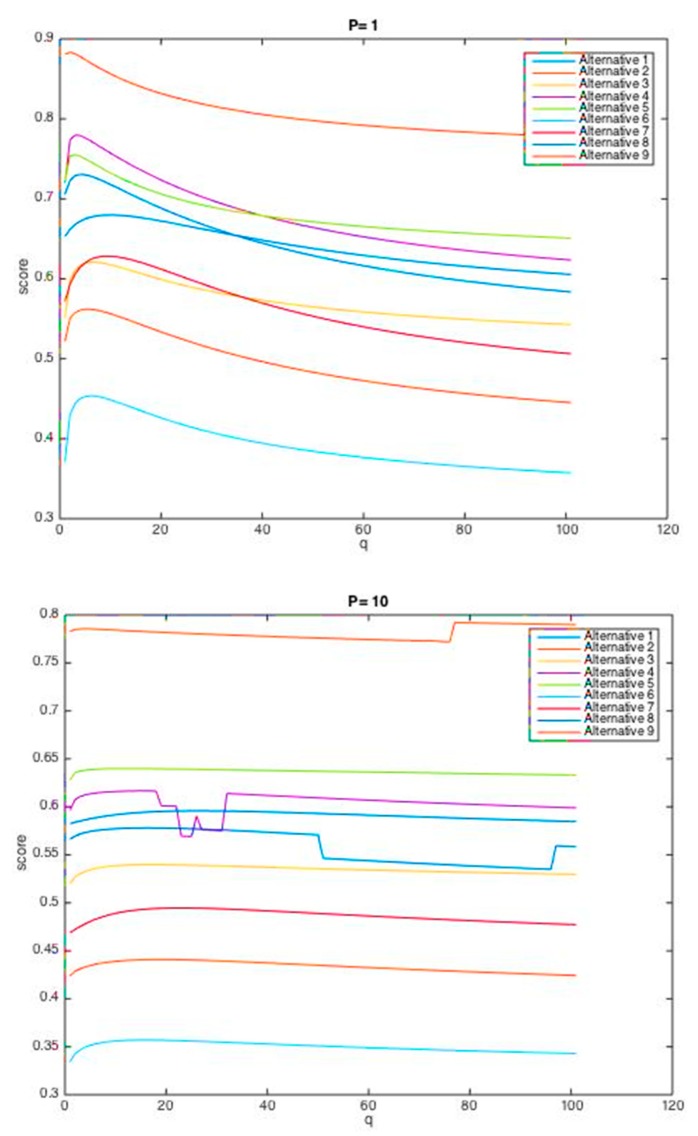
Scores of nine alternatives (q ∈ (0,10]; p = 1, p = 10).

**Figure 3 ijerph-17-01442-f003:**
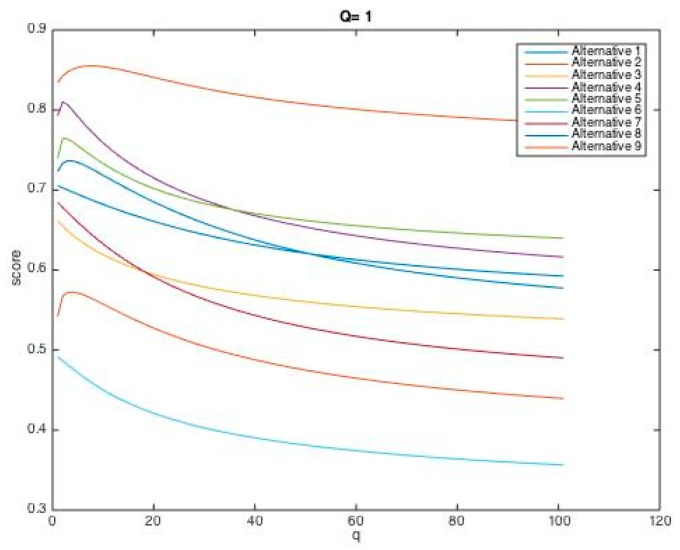
Scores of nine alternatives (p ∈ (0,10]; q = 1, q = 10).

**Figure 4 ijerph-17-01442-f004:**
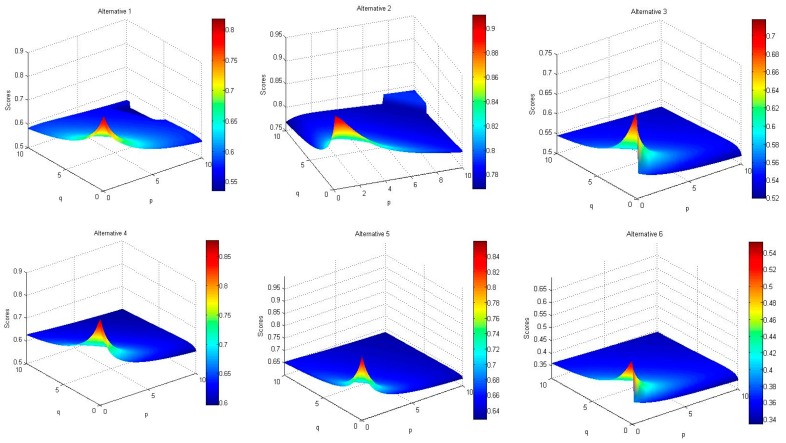
Scores obtained by the HFWHM operator (p ∈ [0,10], q ∈ [0,10]).

**Figure 5 ijerph-17-01442-f005:**
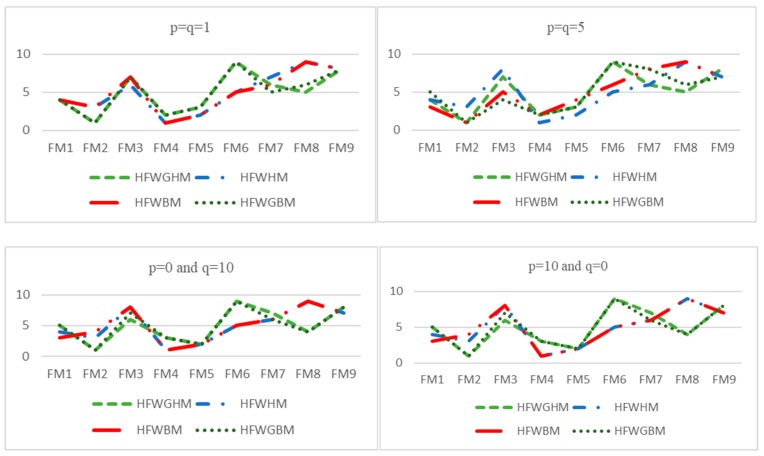
The results of comparison analysis.

**Table 1 ijerph-17-01442-t001:** Hesitant fuzzy decision matrix.

Failure Mode/Risk Factor	$C_{1}\;\left(O\right)$	$C_{2}\;(S)$	$C_{3}\left(D\right)$
$FM_{1}$	$h_{11}$	$h_{12}$	$h_{13}$
$FM_{2}$	$h_{21}$	$h_{22}$	$h_{23}$
….	…	…	…
$FM_{m}$	$h_{m1}$	$h_{m2}$	$h_{m3}$

**Table 2 ijerph-17-01442-t002:** Potential failure modes (Occupational hazards in the construction industry).

Symbol	Failure Mode
FM1	Fall from height
FM2	Falling objects
FM3	Collision with objects
FM4	Explosions and fires
FM5	Electrocution
FM6	Structure collapse
FM7	Poisoning
FM8	Hyperthermia and frostbite
FM9	Sound pollution

**Table 3 ijerph-17-01442-t003:** Results of SWARA method in weighting assessment by FMEA team member 1.

Criterion	Comparative Importance of $\mathrm{Average}\;\mathrm{Value}\;(S_{j})$	Coefficient $k_{j}=S_{j}+1$	Recalculated Weight $w_{j}=\frac{w_{j-1}}{K_{j}}$	Weight $q_{j}=\frac{w_{j}}{\sum w_{j}}$
Severity	1	1	1	0.403144
Detection	0.110447	1.110447	0.900538	0.363046
Occurrence	0.552744	1.552744	0.579965	0.23381

**Table 4 ijerph-17-01442-t004:** Final weights of risk factors.

FMEA Team Members	O	S	D
Member 1	0.616856	0.31167	0.071474
Member 2	0.232433	0.401204	0.366363
Member 3	0.522071	0.402611	0.075318
Member 4	0.274603	0.332734	0.392663
Geometric mean	0.379	0.360	0.167
Normalized weight (final)	0.418	0.397	0.184

**Table 5 ijerph-17-01442-t005:** Linguistic variables.

Linguistic Variable	Symbol	Value
Very High	VH	1
High	H	0.9
Medium High	MH	0.7
Medium	M	0.5
Medium Low	ML	0.3
Low	L	0.1
Very Low	VL	0

**Table 6 ijerph-17-01442-t006:** Initial decision matrix (by each expert’s opinion).

Failure Modes	Expert 1			Expert 2			Expert 3			Expert 4		
	O	S	D	O	S	D	O	S	D	O	S	D
FM1	MH	VH	H	L	H	H	MH	H	H	L	MH	H
FM2	H	MH	VL	H	MH	MH	H	MH	H	H	M	MH
FM3	VL	M	L	MH	M	MH	M	M	MH	M	M	M
FM4	ML	VH	M	L	VH	M	M	VH	H	L	VH	M
FM5	ML	H	H	ML	VH	H	ML	VH	H	ML	VH	H
FM6	VL	H	H	VL	H	M	M	M	VH	VL	H	M
FM7	L	H	M	L	H	VH	ML	M	MH	L	H	VH
FM8	L	M	M	MH	ML	H	ML	M	H	MH	ML	M
FM9	H	L	M	M	M	H	M	ML	H	L	VL	M

**Table 7 ijerph-17-01442-t007:** Hesitant fuzzy decision matrix (linguistic variables form).

Failure Modes	O	S	D
FM1	{MH, L}	{VH, MH}	{H}
FM2	{H}	{MH, M}	{VL, MH, H}
FM3	{VL, MH, M}	{M}	{L, MH, M}
FM4	{ML, L, M}	{VH}	{M, H}
FM5	{ML}	{H, VH}	{H}
FM6	{M, VL}	{M, H}	{H, M, VH}
FM7	{ML, L}	{M, H}	{M, VH, MH}
FM8	{L, MH, ML}	{M, ML}	{H, M}
FM9	{H, M, L}	{L, M, ML, VL}	{M, H}

**Table 8 ijerph-17-01442-t008:** Hesitant fuzzy decision matrix (numeric form).

Failure Modes	O	S	D
FM1	{0.7,0.1}	{1,0.7}	{0.9}
FM2	{0.9}	{0.7, 0.5}	{0, 0.7, 0.9}
FM3	{0,0.7, 0.5}	{0.5}	{0.1, 0.7, 0.5}
FM4	{0.3, 0.1,0.5}	{1}	{0.5, 0.9}
FM5	{0.3}	{0.9, 1}	{0.9}
FM6	{0.5, 0}	{0.5, 0.9}	{0.9, 0.5, 1}
FM7	{0.3, 0.1}	{0.5, 0.9}	{0.5, 1,0.7}
FM8	{0.1, 0.7, 0.3}	{0.5,0.3}	{0.9, 0.5}
FM9	{0.9, 0.5, 0.1}	{0.1, 0.5, 0.3, 0}	{0.5, 0.9}

**Table 9 ijerph-17-01442-t009:** Normalized hesitant fuzzy decision matrix.

Failure Modes	O	S	D
FM1	{0.7,0.1}	{1,0.7}	{0.}
FM2	{0.9}	{0.7, 0.5}	{1, 0.3, 0.1}
FM3	{0,0.7, 0.5}	{0.5}	{0.9, 0.3, 0.5}
FM4	{0.3, 0.1,0.5}	{1}	{0.5, 0.1}
FM5	{0.3}	{0.9, 1}	{0.1}
FM6	{0.5, 0}	{0.5, 0.9}	{0.1, 0.5, 0}
FM7	{0.3, 0.1}	{0.5, 0.9}	{0.5, 0,0.3}
FM8	{0.1, 0.7, 0.3}	{0.5,0.3}	{0.1, 0.5}
FM9	{0.9, 0.5, 0.1}	{0.1, 0.5, 0.3, 0}	{0.5, 0.1}

**Table 10 ijerph-17-01442-t010:** Performance score and rank of each failure mode for p = q = 1.

Failure Modes	FM1	FM2	FM3	FM4	FM5	FM6	FM7	FM8	FM9
Score	0.7148	0.8532	0.6160	0.7534	0.7291	0.4466	0.6277	0.6797	0.5542
Rank	4	1	7	2	3	9	6	5	8

**Table 11 ijerph-17-01442-t011:** Performance score and rank of each failure mode for different values of p and q.

Failure Modes	p = 0, q = 1		p = q= 1		p = q = 5		p = 0, q = 10		p = 10, q = 0	
	Score	Rank	Score	Rank	Score	Rank	Score	Rank	Score	Rank
FM1	0.723122	4	0.714843	4	0.594653	5	0.579211	5	0.566349	5
FM2	0.834288	1	0.853205	1	0.787039	1	0.768274	1	0.783073	1
FM3	0.661109	7	0.615983	7	0.547139	6	0.544187	6	0.51993	6
FM4	0.792478	2	0.753417	2	0.629211	3	0.624975	3	0.596291	3
FM5	0.73924	3	0.729124	3	0.649512	2	0.648942	2	0.627991	2
FM6	0.491569	9	0.446554	9	0.363921	9	0.358481	9	0.334404	9
FM7	0.68462	6	0.627684	6	0.51142	7	0.512346	7	0.468757	7
FM8	0.705247	5	0.679666	5	0.611169	4	0.606705	4	0.582494	4
FM9	0.541866	8	0.554239	8	0.454828	8	0.433807	8	0.423873	8

**Table 12 ijerph-17-01442-t012:** Results comparison.

	Traditional FMEA		Ranking Using MCDM Method Based on Different Operators of Hesitant Fuzzy Numbers for Different Values of p and q															
			p = q = 1				p = q = 5				p = 0, q = 10				p = 10, q = 0			
	RPN	Rank	HFWGHM	HFWHM	HFWBM	HFWGBM	HFWGHM	HFWHM	HFWBM	HFWGBM	HFWGHM	HFWHM	HFWBM	HFWGBM	HFWGHM	HFWHM	HFWBM	HFWGBM
FM1	245	1	4	4	4	4	4	4	3	5	5	4	3	5	5	4	3	5
FM2	210	3	1	3	3	1	1	3	1	1	1	3	4	1	1	3	4	1
FM3	126	7	7	6	7	7	7	8	5	4	6	8	8	7	6	8	8	7
FM4	192	4	2	1	1	2	2	1	2	2	3	1	1	3	3	1	1	3
FM5	224	2	3	2	2	3	3	2	4	3	2	2	2	2	2	2	2	2
FM6	126	7	9	5	5	9	9	5	6	9	9	5	5	9	9	5	5	9
FM7	147	6	6	7	6	5	6	6	8	8	7	6	6	6	7	6	6	6
FM8	150	5	5	9	9	6	5	9	9	6	4	9	9	4	4	9	9	4
FM9	120	8	8	8	8	8	8	7	7	7	8	7	7	8	8	7	7	8

**Table 13 ijerph-17-01442-t013:** The correlation coefficients of the ranking resulted from different hesitant fuzzy operators.

	**p = q = 1**					**p = q = 5**			
	**HFWGHM**	**HFWHM**	**HFWBM**	**HFWGBM**		**HFWGHM**	**HFWHM**	**HFWBM**	**HFWGBM**
**HFWGHM**	1.00	0.67	0.68	0.98	**HFWGHM**	1.00	0.67	0.70	0.87
**HFWHM**		1.00	0.98	0.70	**HFWHM**		1.00	0.80	0.57
**HFWBM**			1.00	0.73	**HFWBM**			1.00	0.80
**HFWGBM**				1.00	**HFWGBM**				1.00
	**p = 0 and q = 10**					**p = 10 and q = 0**			
	**HFWGHM**	**HFWHM**	**HFWBM**	**HFWGBM**		**HFWGHM**	**HFWHM**	**HFWBM**	**HFWGBM**
**HFWGHM**	1.00	0.53	0.47	0.98	**HFWGHM**	1.00	0.53	0.47	0.98
**HFWHM**		1.00	0.98	0.57	**HFWHM**		1.00	0.98	0.57
**HFWBM**			1.00	0.50	**HFWBM**			1.00	0.50
**HFWGBM**				1.00	**HFWGBM**				1.00

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
