# Peer review of "An Extension of the Failure Mode and Effect Analysis with Hesitant Fuzzy Sets to Assess the Occupational Hazards in the Construction Industry"

_ijerph, 2020, doi:10.3390/ijerph17041442_

Round 1
Reviewer 1 Report
The main aim of this paper is also not clearly presented.
Delete lines 112 to 119 from the text. Do not give article organization in the introduction.
Chapters (sections) of the article should be numbered.
There is no results section at work.
There is also no result discussion in the article.
Therefore, the article should be rewritten.
Author Response
|
Comments |
Description |
Page number |
|
English language and style are fine/minor spell check required |
The paper was checked and the detected errors were corrected. |
Whole the paper |
|
The main aim of this paper is also not clearly presented. |
the goals of the paper were mentioned in the introduction and the literature review sections. |
Page 3: lines 100-106 page 5: lines 219-222 |
|
Delete lines 112 to 119 from the text. Do not give article organization in the introduction. |
The mentioned line were deleted. |
References
|
|
Chapters (sections) of the article should be numbered. |
All of the sections were numbered in the paper. |
Whole the paper |
|
There is no results section at work. |
A separate section was allocated to propose and discuss the results of the paper.Two separate sub-sections were added to this part of the paper, which one of them is devoted to analyzing the sensitivity of the proposed method regarding the different values of p and q parameters, and the other section is devoted to comparing the results of the proposed method with other similar operators. Moreover, some graphs and a related table were added to the discussion section. |
Pages 12-18 Figure 5 Table 12 |
|
There is also no result discussion in the article. |

Reviewer 2 Report
Dear authors,
I have reviewed your manuscript. This includes many strengths and some weaknesses that I believe should correct:
- The manuscript, as the journal points out, must have a numbering in its sections.
- The Introduction is good, but I have the impression that some numerical data of the state of health in this sector or undisclosed, according to WHO, should reflect and justify their contribution.
- I understand that the Hesitant fuzzy sets section is related to other work, and is not an original contribution. For this reason, I do not see the reason for their inclusion in the work, but this should only be cited.
- They should improve their conclusions in the sense of making clear the contribution of this manuscript and why it should be read and taken into consideration. That is, you should justify the reason why the status of the issue of this research topic improves.
I hope you can improve it. Kindly regards!
Author Response
|
Comments |
Description |
Page number |
|
The manuscript, as the journal points out, must have a numbering in its sections. |
All of the sections were numbered in the paper. |
Whole the paper |
|
The Introduction is good, but I have the impression that some numerical data of the state of health in this sector or undisclosed, according to WHO, should reflect and justify their contribution. |
Some related statistics about fatal and non-fatal injuries in the construction industry were added to the paper. |
Page 2 |
|
I understand that the Hesitant fuzzy sets section is related to other work, and is not an original contribution. For this reason, I do not see the reason for their inclusion in the work, but this should only be cited. |
Because we wanted to cite the equations in the calculation process, we had to mention the Hesitant fuzzy sets, its steps, and the related calculations. But, to address this comment, we put it as a sub-set in “The proposed hesitant fuzzy FMEA method” section. |
Page 7 |
|
They should improve their conclusions in the sense of making clear the contribution of this manuscript and why it should be read and taken into consideration. That is, you should justify the reason why the status of the issue of this research topic improves. |
Two separate sub-sections were added to this paper, which one of them is devoted to analyzing the sensitivity of the proposed method regarding the different values of p and q parameters, and the other section is devoted to comparing the results of the proposed method with other similar operators. Moreover, some graphs and a related table were added to the discussion section. Furthermore, in the conclusion section, the knowledge gap and the contribution of the paper were remarked again. |
Pages 16-19 Figure 5 Table 12 |

Reviewer 3 Report
I have reviewed the article and come up with the following comments, which may not be in order:
Authors should provide abbreviation of failure mode and effect analysis (FMEA), multi-criteria decision-making (MCDM), Stepwise Weight Assessment Ratio Analysis (SWARA) in the Abstract. Although the paper started with a good structure and reads well, the literature part needs improvement by looking at some more relevant papers. The authors need to take an intensive review for existing knowledge and knowledge gap should be clearly defined within the context of construction industry. Authors need to see if there is acknowledgement section is needed. Conclusion part looks more like a summary, rather than a significant concluding remarks and research implications. Overall, the paper is interesting. However, it needs a bit of work to make it scientifically worth and publishable.
Author Response
|
Comments |
Description |
Page number |
|
Authors should provide abbreviation of failure mode and effect analysis (FMEA), multi-criteria decision-making (MCDM), Stepwise Weight Assessment Ratio Analysis (SWARA) in the Abstract. |
They were corrected in the abstract |
Abstract |
|
Although the paper started with a good structure and reads well, the literature part needs improvement by looking at some more relevant papers. |
Since the primary goal of this paper is to present a new extension of FMEA in order to overcome its shortcomings by using the SWARA method and HFWGHM operator, the structure of the paper was modified. A literature review section was added to the paper to state the previous extensions of FMEA and demonstrate their deficiencies. At the end of this section, two separate paragraphs were added to remark the knowledge gap and re-state the goal of the paper. |
Page 3, 5 |
|
The authors need to take an intensive review for existing knowledge and knowledge gap should be clearly defined within the context of construction industry. |
As mentioned earlier, the aim of this paper is improving the conventional FMEA to overcome its deficiencies and then applying the proposed method in the construction industry. Consequently, gap analysis in the literature review section is related to the previous improvements of the FMEA and their shortcomings. By the way, to address this comment, we tried to analyze some related works in the construction industry and the potential gaps in this sector. |
page 5 |
|
Conclusion part looks more like a summary, rather than a significant concluding remarks and research implications. |
In the conclusion section, the knowledge gap and the shortcomings of the previous extensions of FMEA were explained, and the contributions of the proposed method were highlighted. |
Pages 18, 19 |

Round 2
Reviewer 1 Report
The manuscript has been significantly improved.
Reviewer 2 Report
Dear authors,
I thank you for the review you have made of the manuscript. This one has improved significantly.
Kind regards